

**Exploring the amplied role of HCHO during the wintertime ozone**
**and PM2.5 pollution in a coastal city of southeast China**
Youwei Hong[a,b,c,d,g*], Keran Zhang[a,b,d], Dan Liao[f], Gaojie Chen[a,b,c], Min Zhao[i], Yiling Lin[a,b,e],
Xiaoting Ji[a,b,c], Ke Xu[a,b,g], Yu Wu[a,b,e], Ruilian Yue[e], Gongren Hu[e], Sung-Deuk Choi[h], Likun Xue[i*],
Jinsheng Chen[a,b,c]
[a]Center for Excellence in Regional Atmospheric Environment, Key Lab of Urban Environment and
Health, Institute of Urban Environment, Chinese Academy of Sciences, Xiamen, 361021, China
[b]Fujian Key Laboratory of Atmospheric Ozone Pollution Prevention, Institute of Urban
Environment, Chinese Academy of Sciences, Xiamen, 361021, China
[c]University of Chinese Academy of Sciences, Beijing, 100049, China
[d]College of Resources and Environment, Fujian Agriculture and Forest University, Fuzhou 350002,
China
[e]College of Chemical Engineering, Huaqiao University, Xiamen, 361021, China
[f]College of Environment and Public Health, Xiamen Huaxia University, Xiamen 361024, China
[g]School of Life Sciences, Hebei University, Baoding, 071000, China
[h]Department of Urban and Environmental Engineering, Ulsan National Institute of Science and
Technology, Ulsan, 44919, South Korea
[i]Environment Research Institute, Shandong University, Qingdao, 266237, China
*Corresponding    author    E-mail:    Youwei    Hong    (ywhong@iue.ac.cn);    Likun    Xue
(xuelikun@sdu.edu.cn)



**Abstract:**
To develop the effective strategies for controlling both $PM_{2.5}$ and $O_3$ levels, it is crucial
to understand their synergistic mechanisms, key precursors, and atmospheric
physiochemical processes involved. In this study, a wintertime co-occurring $O_3$ and
$PM_{2.5}$ pollution event in a coastal city in southeast China was investigated based on
high-time resolution measurements of criteria air pollutants, chemical compositions of
$PM_{2.5}$, and $O_3$ precursors, such as NOx, HCHO, and VOCs. The results of this study
revealed a positive correlation between $PM_{2.5}$ and MDA8 $O_3$ concentrations during the
whole periods, suggesting an increase in atmospheric oxidation capacity (AOC) during
the cold seasons. Strong correlations ($R^2$ = 0.415–0.477) were observed between
HCHO, Fe, Mn, and sulfate concentrations, suggesting the influence of catalyzed
oxidation processes in the coastal city. Through an observation-based model (OBM)
analysis coupled with the regional atmospheric chemistry mechanism version 2
(RACM2) and the chemical aqueous-phase radical mechanism version 3.0 (CAPRAM
3.0), we found that high concentrations of precursors ($SO_2$ and HCHO), high relative
humidity, and moderately acidic pH conditions enhanced the heterogeneous formation
of hydroxymethanesulfonate (HMS) in $PM_{2.5}$. Furthermore, by employing the Master
Chemical Mechanism (OBM-MCM), we verified that disabling the HCHO mechanism
could decrease daytime net $O_3$ production rates by reducing the production rates of
$HO_2$+NO. These results were consistent with the daily values of AOC, OH, $HO_2$, and
$RO_2$ concentrations. This study contributes to a better understanding of the significance



of HCHO in photochemical reactions and the formation of secondary aerosols in a
coastal city.
**Key words:** $PM_{2.5}$; $O_3$; formaldehyde; OBM; coastal city

## Introduction

Air pollution, dominated by fine particulate matter ($PM_{2.5}$) and ground-level ozone

($O_3$), is an important global environmental issue linked to climate change and human
health, including cardiovascular and respiratory illnesses and mortality (Xiao et al.,
2022; Vohra et al., 2022). To decrease global air pollution and associated mortality, the
World Health Organization recently updated its air quality guideline for annual $PM_{2.5}$
exposure from 10 to 5 μg m$^{-3}$ and added the average $O_3$ concentrations of no more than
60 μg m$^{-3}$ during the peak season (WHO, 2021). To develop two-pollutant control
strategies to decrease both $PM_{2.5}$ and $O_3$, there is a need to understand the synergistic
mechanisms and spatiotemporal delineation between them (Ivatt et al., 2022; Li et al.,
2019b).

There are complex synergistic effects between $PM_{2.5}$ and $O_3$, due to common

precursors (e.g., NOx and VOCs), atmospheric physiochemical processes, and weather
systems (Li et al., 2019a; Shao et al., 2022; Jia et al., 2023; Zhang et al., 2022; Qin et
al., 2021; Qu et al., 2023). Some studies have reported that $O_3$ could enhance the
formation of secondary $PM_{2.5}$ by strengthening atmospheric oxidation capacity (Qin et
al., 2022; Zhao et al., 2020). An increase in $O_3$ concentration will increase oxidizing





substances, such as OH, $H_2O_2$, and RCHO, which promote the oxidation of $SO_2$, NOx,
and VOCs to secondary inorganic and organic components in $PM_{2.5}$ (Feng et al., 2020;
Lu et al., 2019). On the contrary, $PM_{2.5}$ could affect $O_3$ formation by interfering with
the radiation intensity of the Earth, providing a multiphase reaction surface and
affecting the radiation flux and intensity of the boundary layer (Shao et al., 2021; Li et
al., 2017). In addition, multiphase reactions occur on the surface of atmospheric
particles, such as the hydrolysis of $N_2O_5$ and $HO_2$ absorption, thus affecting the
formation of $O_3$ precursors $NO_2$ (Song et al., 2022; Lou et al., 2014).

Formaldehyde (HCHO) plays an important role in the photochemical reaction

process and secondary aerosol formation (Kalashnikov et al., 2022; Ma et al., 2020;
Song et al., 2021; Zong et al., 2021). Most studies have focused on pollution
characteristics and sources, particle uptake of HCHO, and their impacts on atmospheric
oxidation capacity (Liu et al., 2022b; Wu et al., 2023; Zhang et al., 2021). Recent
studies have reported that atmospheric HCHO contributes to sulfate formation in $PM_{2.5}$
by producing $HO_2$ radicals and hydroxymethyl hydroperoxide (HMHP) or
hydroxymethanesulfonate (HMS) (Wu et al., 2023; Dovrou et al., 2022; Campbell et
al., 2022). These studies highlight the necessity for more observation research to obtain
evidence of the contributions of HCHO to HMS formation. However, in subtropical
coastal regions with apparent HCHO production, further studies are required to
investigate the impacts of HCHO on the synergistic effects between $PM_{2.5}$ and $O_3$.

Xiamen, a coastal city in southeast China, has frequently experienced $PM_{2.5}$

pollution (with low $O_3$ concentrations) in winter and $O_3$ pollution in spring and autumn



(Hong et al., 2022; Wu et al., 2019). Our previous studies mainly focused on the
pollution characteristics of PM$_{2.5}$ or O$_3$ in different seasons and their sources associated
with anthropogenic emissions and the East Asian monsoon (Liu et al., 2020; Liu et al.,
2022a; Hong et al., 2021). At the end of winter and the beginning of spring in 2022, an
outbreak of co-occurring O$_3$ and PM$_{2.5}$ pollution was observed in Xiamen. Therefore,
it provided a unique opportunity to study the interactions among precursors,
heterogeneous chemistry, and photochemical reactions for the synergistic effects of
PM$_{2.5}$ and O$_3$. In the coastal region, there is an apparent alternation of polluted and clean
air masses from continental and ocean areas and a local geographical environment,
including relatively high humidity, dense vegetation, and strong atmospheric oxidation
capacity (Hu et al., 2022; Wu et al., 2020). Potential synergistic mechanisms between
O$_3$ and PM$_{2.5}$ would differ from those in megacities of China, such as the Beijing-
Tianjin-Hebei (BTH) Area, the Yangtze River Delta (YRD), and the Pearl River Delta
(PRD). Based on the observation-based model (OBM) analysis, the objectives of this
study are to (1) characterize the wintertime co-occurring O$_3$ and PM$_{2.5}$ pollution process
in a coastal city; (2) elaborate the influence of HCHO on the heterogeneous formation
of hydroxymethanesulfonate (HMS) in PM$_{2.5}$; (3) explore the mechanisms of HCHO
on O$_3$ pollution and photochemical reactions process.

**2 Methods and materials**
**2.1 Study area**





The monitoring site (Institute of Urban Environment, Chinese Academy of
Sciences, 118.06° E, 24.61° N) is located in Xiamen, a coastal city in southeast China
(Fig. S1). It is situated in a subtropical monsoon climate, with an annual average
temperature of 23.3°C and a relative humidity of 77.6%. In autumn and winter, cold
and dry air masses move northward from inland, while in late spring and summer, the
prevailing air masses are southerly, characterized by warm air temperatures and high
humidity. The air-monitoring supersite is located on the rooftop of a building,
surrounded by residential buildings, educational institutions, a commercial zone, and
freeways. The downtown area of Xiamen, with a high population density and frequent
traffic jams, is located south of the monitoring site.

**2.2 Observation**
Gas and aerosol species, $O_3$ precursors, photolysis rate, and meteorological
parameters were continuously measured online from February 17 to March 17, 2022.
Hourly mass concentrations of $PM_{2.5}$ and $PM_{10}$ were measured using a tapered element
oscillating microbalance (TEOM1405, Thermo Scientific Corp., MA, USA). $NO/NO_2$,
$SO_2$, and $O_3$ were monitored using continuous gas analyzers (TEI 42$i$, 43$i$, and 49$i$,
Thermo Scientific Corp., MA, USA). HCHO analyzer (FMS-100, Focused Photonics
Inc., Hangzhou, China) was used to measure gaseous HCHO based on the Hantzsch
reaction, according to our previous method (Liu et al., 2022b). Water-soluble inorganic
ions (WSII) in $PM_{2.5}$ ($Cl^-$, $SO_4^{2-}$, $NO_3^-$, $Na^+$, $K^+$, $NH_4^+$, $Mg^{2+}$, and $Ca^{2+}$) were measured
hourly using a Monitoring device for AeRosols and Gases in ambient Air (MARGA



2080; Metrohm Applikon B.V.; Delft, Netherlands). Simultaneously, organic carbon
(OC) and elemental carbon (EC) in $PM_{2.5}$ were measured using an OC/EC analyzer
(model RT-4; Sunset Laboratory Inc.; Tigard, USA). BC was monitored using an
Aethalometer (AE31, Magee Scientific, USA) with a $PM_{2.5}$ cut-off inlet. Besides,
concentrations of 22 elements (Al, Si, S, Fe, K, Mn, Pb, Ca, Zn, Ba, V, Cu, Ni, As, Cr,
Ag, Se, Br, Hg, Sn, Ti, and Sb) were measured using a multi-metal monitor (Xact™
625, Cooper Environmental Services, LLT; Portland, USA). Strict quality assurance
and quality control procedures were applied, and the maintenance and accuracy of all
online instruments were validated (Hong et al., 2021).

A gas chromatograph-mass spectrometer (GC-FID/MS, TH-300B, Wuhan, China)

was used to measure ambient VOCs with one-hour time resolution, following the
method from our previous studies (Liu et al., 2020a,b). Briefly, the air sample was
preconcentrated by cooling to −160 °C in a cryogenic trap, then heated to 100 °C, and
subsequently transferred to the secondary trap using high-purity helium (He). The low-
carbon (C2-C5) hydrocarbons were detected using a flame ionization detector (FID)
with a PLOT ($Al_2O_3$/KCl) column (15 m × 0.32 mm × 6.0 μm), while other VOC
species were quantified using a GC/MS with a DB-624 column (60 m × 0.25 mm × 1.4
μm). The instrument can quantify 106 VOC species, including 29 alkanes, 11 alkenes,
one alkyne, 17 aromatics, 35 halogenated hydrocarbons, and 13 OVOCs. Calibration
was performed daily at 23:00 using the standard mixtures of US EPA PAMS and TO-
15. The detection limits of the measured VOCs ranged from 0.02 ppbv to 0.30 ppbv.

Ambient meteorological parameters, including relative humidity (RH),



temperature (T), wind speed (WS), and wind direction (WD), were obtained using an
ultrasonic atmospherium (150WX, Airmar, USA). Photolysis frequencies and HCHO
were measured using a photolysis spectrometer PFS-100 and a formaldehyde monitor
FMS-100 (Focused Photonics Inc., Hangzhou, China), respectively. The photolysis rate
constants include $J(O^1D)$, $J(NO_2)$, $J(H_2O_2)$, $J(HONO)$, $J(HCHO)$, and $J(NO_3)$. The
distribution of fire spots during the observation periods was obtained from the Fire
Information        for        Resource        Management        System
(https://firms.modaps.eosdis.nasa.gov/firemap/). The data for boundary layer height
(BLH) were obtained from the European Centre for Medium-Range Weather Forecasts
(ECMWF)        ERA5        hourly        reanalysis        dataset
(https://www.ecmwf.int/en/forecasts/datasets/reanalysis-datasets/era5,    last    access:
March 24, 2023).

**2.3 Positive matrix factorization (PMF) analysis**
The PMF 5.0 model was applied to quantify high-time-resolution sources of PM$_{2.5}$
during the observation periods. The details of the model analysis were described in our
previous studies (Hong et al., 2021; Liu et al., 2020). Briefly, Eq. (1) demonstrates $j$
compound species in the $i$th sample as the concentration from $p$ independent sources.

$$x_{ij} = \sum_{k=1}^{p} g_{ik} f_{ki} + e_{ij}$$

(1)

Where $e_{ij}$ is the residual for each species, $f_{kj}$ is the fraction of the $j$th species from
the $k$th source, $g_{ik}$ is the species contribution of the $k$th source to the $i$th sample, $x_{ij}$ is





the $j$th species concentration measured in the $i$th sample, and $p$ is the total number of
independent sources. The Q value (Eq. (2)), based on the uncertainties ($\mu$), was used to
evaluate the steadiness of the solution.

$$Q = \sum_{i=1}^{n} \sum_{j=1}^{m} \left[ \frac{x_{ij} - \sum_{k=1}^{p} g_{ik} f_{kj}}{\mu_{ij}} \right]^2$$

(2)


**2.4 Observation-based model (OBM)**


The OBM-MCM model is employed to simulate in situ atmospheric

photochemical processes and quantify the $O_3$ production rate, AOC, and OH reactivity.
The details of the OBM-MCM model were reported in our previous studies (Liu et al.,
2022a,b). In summary, monitoring data with a one-hour time resolution of air pollutants
(i.e., $O_3$, CO, NO, $NO_2$, HONO, $SO_2$, and VOCs), meteorological parameters (i.e., T, P,
and RH), and photolysis rate constants ($J(O^1D)$, $J(NO_2)$, $J(H_2O_2)$, $J(HONO)$, $J(HCHO)$,
and $J(NO_3)$) were input into the OBM-MCM model as constraints for the model
simulation. The photolysis rates of other molecules were determined by solar zenith
angle and scaled using measured $J$NO$_2$ values (Saunders et al., 2003). The model
incorporates the physical process of deposition within the boundary layer height (BLH),
which varies from 300 m during nighttime to 1500 m during the daytime in autumn (Li
et al., 2018). Therefore, dry deposition velocities were used to simulate the deposition
loss of certain reactants in the atmosphere (Zhang et al., 2003; Xue et al., 2014).

To simulate the concentration of particulate HCHO and its role in the

heterogeneous formation of hydroxymethanesulfonate (HMS), we combined the OBM



model with the regional atmospheric chemistry mechanism version 2 (RACM2) and
the chemical aqueous-phase radical mechanism version 3.0 (CAPRAM 3.0). We also
considered the mass transfer processes between the gas and aqueous phases (Schwartz,
1986). The Henry's law constant of HCHO is $0.31 \times 10^8$ M atm$^{-1}$, as estimated by
Mitsuishi et al. (2018). For the aqueous HMS formation mechanisms, dissolved HCHO
reacts with sulfite and bisulfite to form HMS (Eq (3-4)), which can be further oxidized
by aqueous OH radicals (Eq (5)).
$$HCHO_{(aq)} + HSO_3{}^- = HOCH_2SO_3{}^- \qquad\qquad\qquad (3)$$
$$HCHO_{(aq)} + SO_3{}^{2-} + H_2O = HOCH_2SO_3{}^- + OH^- \qquad\qquad (4)$$
$$HOCH_2SO_3{}^- + OH \cdot = 2SO_4{}^{2-} + HCHO_{(aq)} + 3H^+ \qquad\qquad (5)$$
The observation data of gaseous NO, NO$_2$, O$_3$, SO$_2$, CO, HCHO, VOCs,
particulate phase NO$_3^-$, NH$_4^+$, and Cl$^-$, along with meteorological parameters with a 1-
h time resolution were interpolated to constrain the model, while the measured SO$_4^{2-}$
was used as the initial condition for the model simulation. Liquid water content (LWC)
and aqueous H$^+$ concentrations, calculated using the ISORROPIA-II model (Hong et
al., 2022), were also used to constrain the model. Model calculations were conducted
from February 26 to March 16, 2022. For each case, the model was initiated at 00:00
local time (LT), and the integration had a step of 1 h and a duration of 24 h.

**2.5 Backward trajectory analysis**
Hybrid Single-Particle Lagrangian Integrated Trajectory (HYSPLIT) was used to



analyze the air masses before and during the $PM_{2.5}$ and $O_3$ pollution period. The 72-h
backward trajectories at a height of 100 m obtained from the National Oceanic and
Atmospheric Administration were run every hour. Cluster analysis was performed, and
four clusters were determined based on the total spatial variance (TSV).

**3 Results and discussions**
**3.1 Overview of co-occurring $O_3$ and $PM_{2.5}$ pollution**

The time series of criteria air pollutants, $O_3$ precursors, and meteorological

parameters from February 17 to March 17, 2022 are shown in Fig. 1. Two typical $PM_{2.5}$
and $O_3$ pollution episodes (EP1: February 26 to March 5; EP2: March 11 to March 17)
were observed, compared to the other periods (Pre-EP1: February 11 to February 25
and Pre-EP2: March 16 to March 10) affected by rainfall. The mean concentrations of
$PM_{2.5}$ during EP1 and EP2 were 51.9 μg m$^{-3}$ and 35.3 μg m$^{-3}$, respectively, compared
to 9.03 μg m$^{-3}$ during Pre-EP1 (Table S1). The concentrations of other air pollutants,
such as $O_3$, $SO_2$, $NO_2$, $PM_{10}$, OC, EC, BC, HCHO, and VOCs, showed a significant
increasing trend during EP1 and EP2. The maximum $PM_{2.5}$ and $O_3$ concentrations were
approximately 100 μg m$^{-3}$ and 200 μg m$^{-3}$, respectively. The maximum daily 8 h
average (MDA8) $O_3$ concentrations were calculated according to the Ambient Air
Quality Standard of China. Fig. S2 shows the positive correlation between $PM_{2.5}$ and
MDA8 $O_3$ concentrations during the whole period. In Xiamen, a coastal city in
Southeast China, the annual mean concentrations of criteria air pollutants from 2015 to



2021 were significantly lower than in other Chinese cities (Fig. S3) (Li et al., 2022;
Shao et al., 2022). Therefore, these two typical PM$_{2.5}$ and O$_3$ pollution episodes (EP1
and EP2) might be worth exploring in terms of the formation mechanisms and
synergistic effects of PM$_{2.5}$ and O$_3$ in the coastal city.

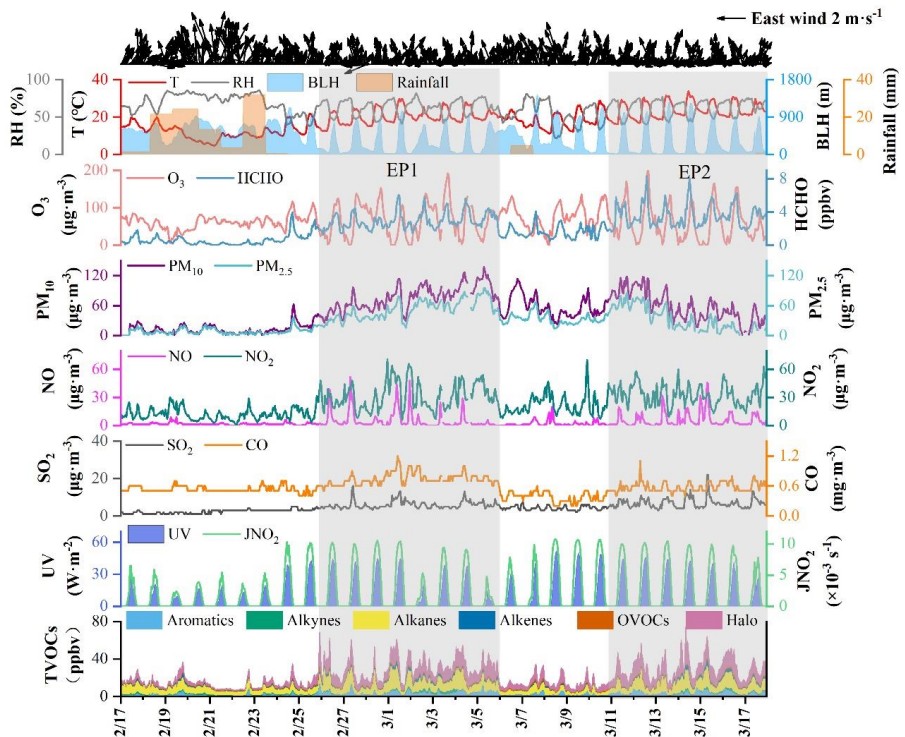


**Fig. 1. Time series of various air pollutants and meteorological parameters**


**3.2 Chemical compositions and sources of PM$_{2.5}$**

As shown in Figure 2, an overall increasing trend is clearly observed in both bulk

PM$_{2.5}$ and its major components during EP1 and EP2. Nitrate and organic matter (OM)
remain the top two dominant PM$_{2.5}$ components, followed by sulfate and ammonium.





The mean concentrations of $SO_4^{2-}$, $NO_3^-$, and $NH_4^+$ during EP1 and EP2 were 7.07 µg
m$^{-3}$ and 5.87 µg m$^{-3}$, 14.95 µg m$^{-3}$ and 9.69 µg m$^{-3}$, and 6.77 µg m$^{-3}$ and 4.46 µg m$^{-3}$,
respectively (Table S1). The increase in EC indicates the contributions of local
anthropogenic emission sources, such as vehicle exhausts (Fig. 2). The concentrations
and percentages of OC and EC during different periods are illustrated in Fig. S4 and
Table S1. The average OC and EC concentrations during EP1 and EP2 were 6.36 µg
m$^{-3}$ and 7.48 µg m$^{-3}$ and 1.23 µg m$^{-3}$ and 1.29 µg m$^{-3}$, respectively, which were notably
higher than those during Pre-EP1 and Pre-EP2. These results are consistent with the
increase in primary emissions and secondary formation contributing to complex air
pollution during the rapid urbanization and industrialization stages in China (Xiao et
al., 2022; Jiang et al., 2022).

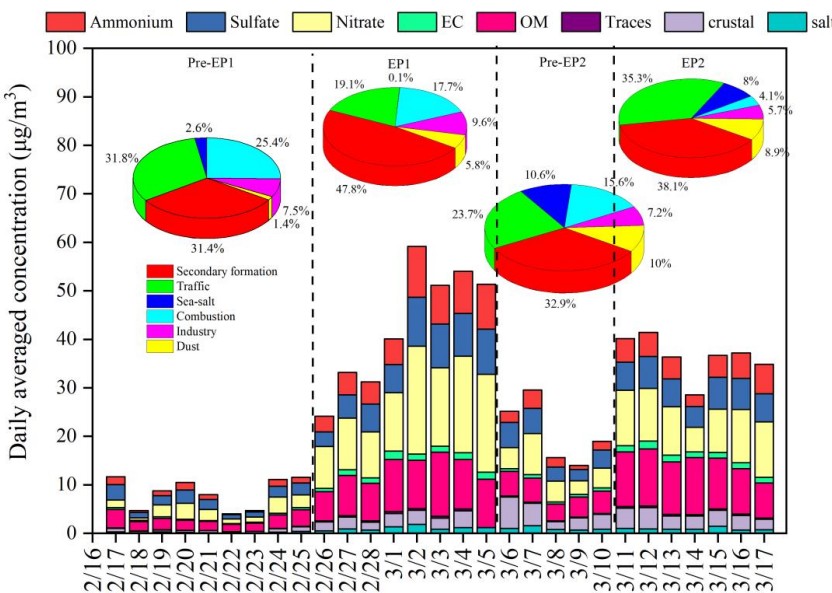


**Fig. 2. Time series of PM$_{2.5}$ chemical composition and sources apportionment by the PMF**
**model. In the legends, OM refers to organic matter, calculated as 1.4×OC; "Tracers" includes**
**elements other than Na, Cl, S, K, Al, Si, Ca, Fe; "Crustal" represents crustal materials,**



**calculated as 1.89×Al+2.14×Si+1.4×Ca+1.43×Fe; and "Salts" includes Na$^+$ and Cl$^-$.**
The PMF model was applied to conduct high-time-resolution source
apportionment of PM$_{2.5}$, based on online hourly measurement data (Hong et al., 2021;
Chow et al., 2022). The factor profiles and the contributions of various sources to PM$_{2.5}$
are shown in Fig. 2 and Fig. S5. Previous studies have indicated that construction and
road dust is characterized by high loadings of Al, Si, Ca$^{2+}$, Na$^+$, Mg$^{2+}$, and Zn (Rienda
and Alves, 2021). In this study, the factor of dust (Factor 1) was identified by the high
contributions of Si (Fig. S5). The PMF analysis revealed that the contribution of dust
to PM$_{2.5}$ ranged from 5.8% to 8.9% during EP1 and EP2, compared to 1.4% during Pre-
EP1 (Fig.2). Factor 2, contributing to the high loading of metal elements (Mn, Zn, Fe,
Pb, and As), was characterized by industrial emissions (Belis et al., 2019). The
contributions from the industry during EP1 and EP2 remained constant. In factor 3, K$^+$
was dominant, and it was identified as coming from combustion sources (Watson et al.,
2001). Biomass burning could change the contribution of combustion to PM$_{2.5}$ at the
monitoring site through long-range transport. During EP2, the influence of combustion
sources (e.g., biomass burning) significantly decreased, due to reduced anthropogenic
emissions and the arrival of clean air masses from the ocean (Figs. S6 and S7). Factor
4, with the highest proportion of Na$^+$ and Mg$^{2+}$ loadings, was associated with the
influence of sea-salt aerosol (Polissar et al., 1998). The percentages of sea salt (8-10%)
during Pre-EP2 and EP2 were relatively high. Factor 5 exhibited high contributions of
EC, OM, and Pb, which are general indicators of vehicle exhaust (Belis et al., 2019).
During EP2, the contribution of traffic increased up to 35.3%. Factor 6 was associated



with secondary aerosol, characterized by high loadings of $SO_4^{2-}$, $NO_3^-$, and $NH_4^+$. The
increased contributions of secondary formation during EP1 and EP2 accounted for 47.8%
and 38.1%, respectively.

**3.3 Formation mechanism of $PM_{2.5}$**

As shown in Fig. S8, $SO_4^{2-}$ was correlated with $NH_4^+$ ($R^2$ = 0.72–0.88), and the

line fit of $NH_4^+$ and $SO_4^{2-}$ showed a slope of 1.78-2.67, suggesting the dominant form
of $(NH_4)_2SO_4$. Similarly, $NO_3^-$ was also correlated with $NH_4^+$ ($R^2$ = 0.77–0.93),
indicating the presence of $NH_4NO_3$. In addition, the ratio of $NH_4^+$ to the sum of $NO_3^-$
and $SO_4^{2-}$ was close to 1, indicating complete neutralization of sulfate and nitrate by
ammonium (Fig. 8c). However, there was no significant difference in the existing form
of SNA in $PM_{2.5}$ under different periods.

The variations of SOR and NOR under different periods are shown in Table S1. It

should be noted that SOR (0.38±0.18) and NOR (0.32±0.08) during EP1 were the
highest, indicating a high oxidation rate of $SO_2$ and $NO_2$. According to RH, T, and UV
(Table S1), noticeable differences in meteorological conditions were observed under
different periods. In this study, LWC was positively correlated with $SO_4^{2-}$, $NO_3^-$, and
$NH_4^+$ (known as the secondary inorganic aerosol, SIA) (Fig. S9), suggesting the
influence of the aqueous phase process, including reactions with $O_3$, OH, $H_2O_2$, and
organic peroxides (Gen et al., 2019; Wang et al., 2023).

Current studies have found that $O_3$, $H_2O_2$, OH, and transition-metal-catalyzed

(TMI) $O_2$ can trigger the secondary formation of $SO_4^{2-}$ (Hong et al., 2021; Gen et al.,





2019). However, the relative importance of these oxidants in enhancing the formation
of $SO_4^{2-}$ is still a topic of debate. As shown in Fig. 3(b) and (c), a good correlation was
found between $SO_4^{2-}$ and Fe and Mn. The TMI-catalyzed oxidation contributed to the
formation of $SO_4^{2-}$, which occurred in both cloud processes and during haze episodes
(Li et al., 2020) because the Mn catalytic reaction rapidly occurred at the aerosol surface
and could oxidize S(IV) through the production of intermediate Mn(III) (Wang et al.,
2021). Even at very low concentrations of Mn, the Mn catalytic reaction, consuming
oxygen and $SO_2$, could produce sulfate. In addition, as an important intermediate
product in atmospheric photochemical reactions, the formation and removal of HCHO
are closely related to OH and $HO_2$ radicals, which directly affect atmospheric reactivity
and oxidation ability (Wu et al., 2023; Zhang et al., 2021). In this study, the correlations
($R^2 = 0.415$) between HCHO and sulfate concentrations were also examined, as
displayed in Fig. 3(a). Recent studies have shown that HCHO can react with hydrogen
peroxide ($H_2O_2$) to produce hydroxymethyl hydroperoxide, which rapidly oxidizes
dissolved sulfur dioxide ($SO_2$, aq) to sulfate (Dovrou et al., 2022). Meanwhile, HCHO
reacts with dissolved $SO_2$ (aq) to produce hydroxymethanesulfonate (HMS), which,
upon oxidation with the hydroxyl radical (OH), forms sulfate (Ma et al., 2020; Moch et
al., 2020).





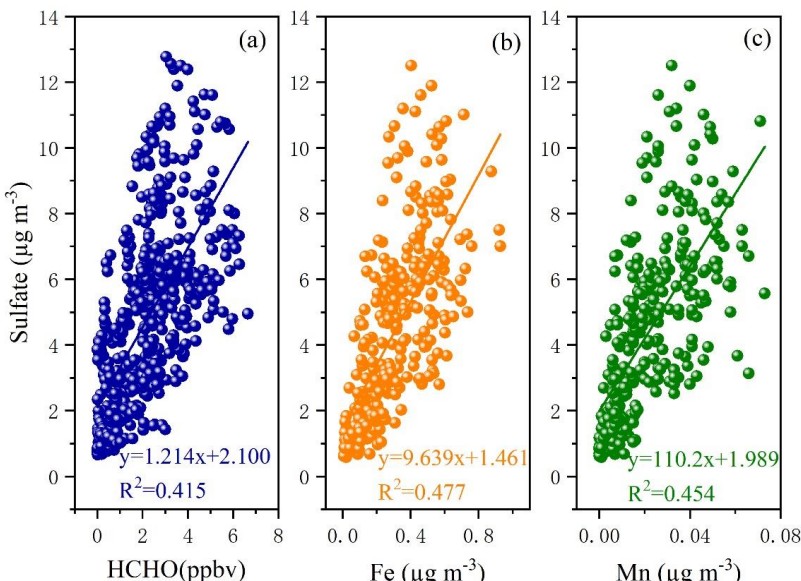

Fig.3. Correlations between the concentrations of sulfate and HCHO (a), Fe (b), and Mn (c)

**3.4 Effects of HCHO on HMS in PM$_{2.5}$**

To calculate the particulate concentrations of HCOH and its contributions to the heterogeneous formation of HMS, we combined the OBM model with RACM2 and CAPRAM 3.0. During EP1 and EP2, the concentrations of HCHO (aq) and HMS (aq), as well as the particulate sulfur molar percentage, increased with the rise of SO$_2$, SO$_4^{2-}$, and HCHO concentrations (Fig. 4). The increase in sulfate concentration was assocaited with the increase in LWC (Fig. S9). Previous studies indicated that the pH and liquid water content of aerosols were the main factors influencing the HCHO uptake coefficient (γ). Moreover, γ has a strong positive exponential relationship with aqueous sulfate concentration (Xu et al., 2022).

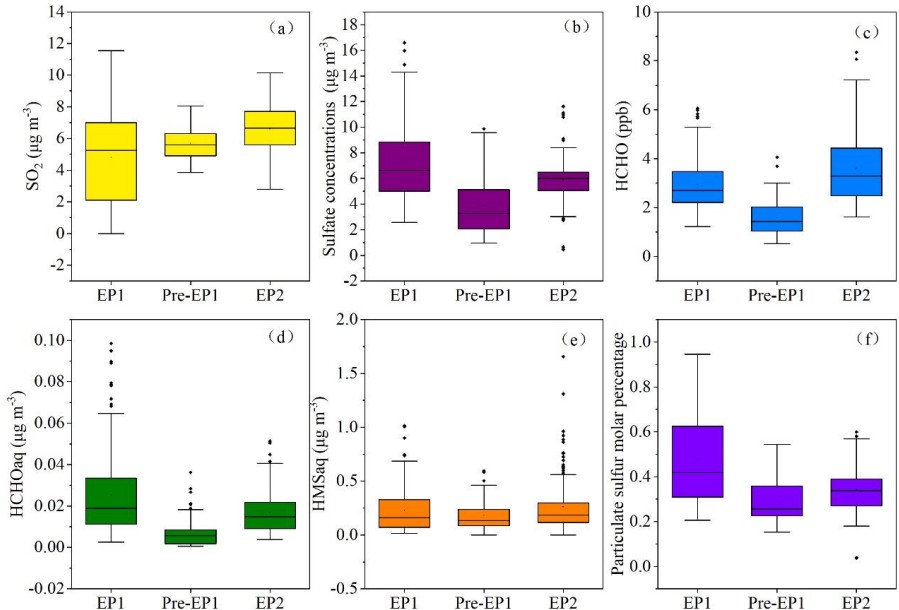

**Figure 4. Concentrations of SO₂, SO₄²⁻, HCHO, HCHO (aq), HMS (aq), and particulate sulfur molar percentage at different pollution levels. In the box–whisker plots, the whiskers, boxes, and points indicate the 5th/95th, 25th/75th, 50th percentiles, and mean values. The particulate sulfur molar percentage was calculated as [n(SO₄²⁻)+n(HMS)]/[n(SO₄²⁻)+n(HMS)+ n(SO₂)].**

As shown in Fig. 5, the concentration of HMS exhibited a similar diurnal variation to that of HCHO (aq). These findings are consistent with the fact that HMS is formed throuhg the reaction between dissolved SO₂ and formaldehyde (HCHO) in aerosol liquid water. In our previous studies, we osbserved that gaseous HCHO showed an increasing trend after sunrise, peaking at noon due to photochemical reactions (Liu et al., 2022b). However, during EP1 and EP2, high concentrations of HCHO (aq) were observed during nighttime. Meanwhile, the heterogeneous formation of HMS also occurred, resulting in elevated HMS concentrations during nighttime.



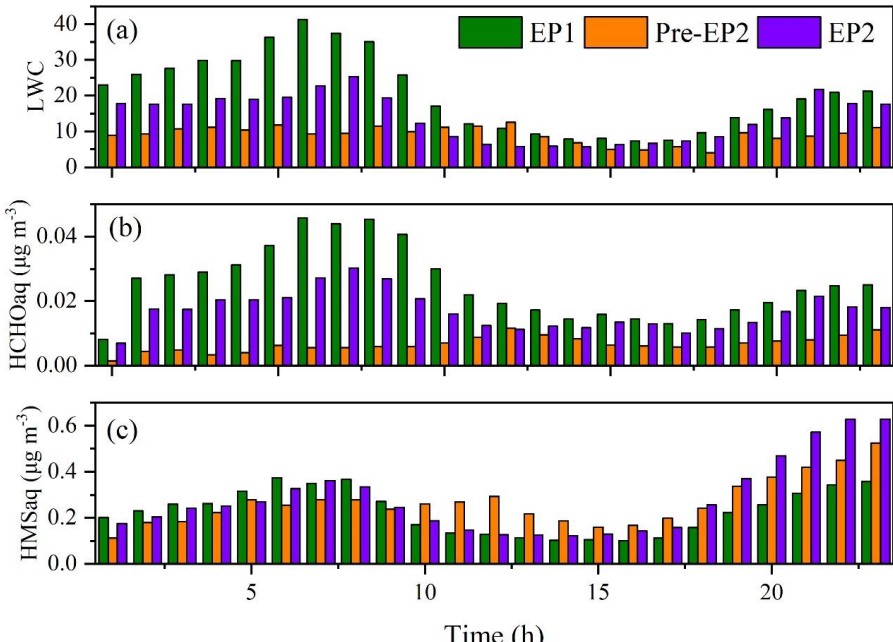

**Figure 5. Diurnal variations of calculated HCHO (aq) and HMS (aq) at different pollution levels**

In this study, high concentrations of HMS were observed under high RH and moderately acidic pH conditions (Fig. 6). Previous studies have also indicated that high RH promotes rapid HMS formation during winter haze, as the aerosol water content could provide numerous reaction interfaces for HMS formation (Ma et al., 2020). Meanwhile, atmospheric sulfur tended to distribute into the particle phase with increasing RH. Fig. 6 shows that HMS formation is favored under pH conditions close to 4.0. Previous studies reported that high HMS concentrations were found under moderate-pH conditions, as low pH inhibits HMS formation, and high pH is unsuitable for its preservation (Ma et al., 2020; Campbell et al., 2022). Therefore, the combination of high precursor concentrations ($SO_2$ and HCHO), high RH, and moderately acidic pH enhanced the heterogeneous formation of HMS in this coastal city.



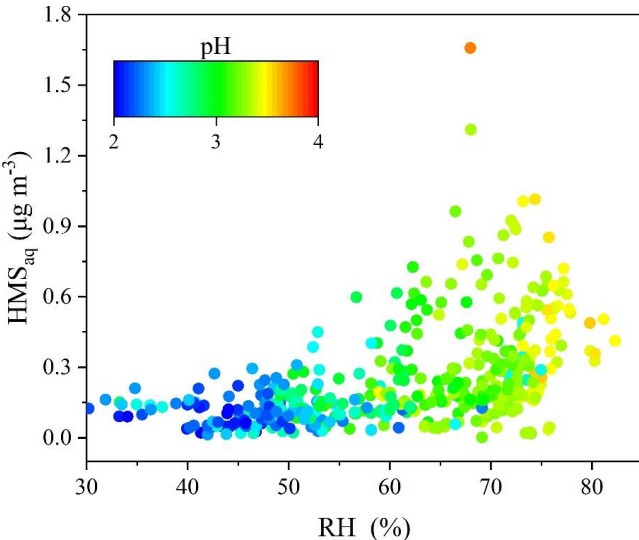

**Figure 6. Evolution of HMS (aq) distribution with increasing RH, colored according to aerosol pH.**

## 3.5 Effects of HCHO on $O_3$ formation

To investigate the effects of HCHO on $O_3$ formation during the co-occurring $O_3$ and $PM_{2.5}$ pollution period, the OBB was used to quantify the detailed $O_3$ production and loss pathways in both scenarios: input HCHO (IH) and non-input HCHO (NIH) (Fig. 7 and Fig. S10). The daytime production rates of $HO_2+NO$ and $RO_2+NO$ in the IH scenario were calculated to be 6.84 and 1.25 ppbv $h^{-1}$ for EP1 and 9.91 and 2.17 ppbv $h^{-1}$ for EP2, respectively. Meanwhile, the predominant $O_3$ loss reaction in this scenario was $OH+NO_2$, with rates of 2.26 ppbv $h^{-1}$ for EP1 and 3.17 ppbv $h^{-1}$ for EP2, followed by $O_3$ photolysis with rates of 0.77 ppbv $h^{-1}$ and 1.10 ppbv $h^{-1}$. In contrast, the daytime production rates of $HO_2+NO$ and $RO_2+NO$ in the NIH scenario were 4.03 and 0.85 ppbv $h^{-1}$ for EP1 and 4.86 and 1.29 ppbv $h^{-1}$ for EP2, respectively. These



results indicate that disabling the HCHO mechanism reduced the production rates of
$HO_2$+NO by 41% for EP1 and 51% for EP2. In addition, the average maximum net $O_3$
production rate observed with the IH scenario was 5.02 ppb h$^{-1}$ for EP1 and 7.93 ppb
h$^{-1}$ for EP2, approximately two times higher than the values of 2.48 ppb h$^{-1}$ and 3.14
ppb h$^{-1}$ observed with the NIH scenario. The results showed that the daytime net $O_3$
production rates decreased by 50–60% when the HCHO mechanism was disabled,
probably due to the decrease in ROx concentrations and radical propagation rates (Wu
et al., 2023; Zhang et al., 2021).

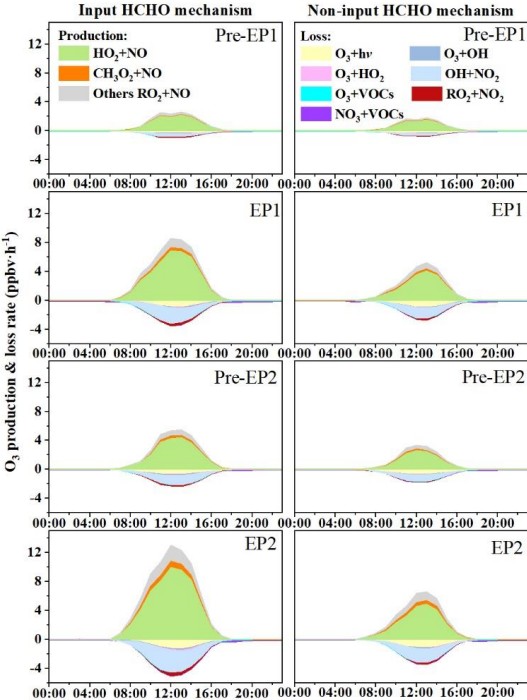


**Figure 7. $O_3$ production and loss rates by the OBM with and without the HCHO mechanism**
The atmospheric oxidation capacity (AOC) is a critical factor in determining the
production rate of secondary pollutants and atmospheric photochemical pollution (Jia



et al., 2023; Qin et al., 2022). In this study, AOC is calculated as the sum of the oxidation
rates of various primary pollutants (e.g., CO, NOx, and VOCs) by major oxidants (i.e.,
OH, $O_3$, and $NO_3$). The model-simulated AOC, OH, $HO_2$, and $RO_2$ under different
periods are shown in Fig. 8 and Fig. 9. The daily maximum AOC during EP1 and EP2
was $8.24\times10^7$ and $11.6\times10^7$ molecules $cm^{-3}$ $s^{-1}$, respectively, which were higher than
those ($2.56\times10^7$ and $5.39\times10^7$ molecules $cm^{-3}$ $s^{-1}$) in other periods. However, when the
HCHO mechanism was disabled, the daily maximum AOC during different stages
decreased significantly. Especially, HCHO played much important role in AOC during
the co-occuring $PM_{2.5}$ and $O_3$ pollution periods. All these results are comparable to rural
sites in Hong Kong ($6.2 \times 10^7$) and Berlin ($1.4\times 10^7$ molecules $cm^{-3}$ $s^{-1}$) but lower than
those observed in highly polluted cities, such as Santiago ($3.2\times10^8$ molecules $cm^{-3}$ $s^{-1}$)
and Shanghai ($1.0\times10^8$ molecules $cm^{-3}$ $s^{-1}$) (Li et al., 2018; Xue et al., 2016; Liu et al.,
2022a). These studies have reported that the variations in AOC are related to precursor
concentrations/types and photochemical conditions.



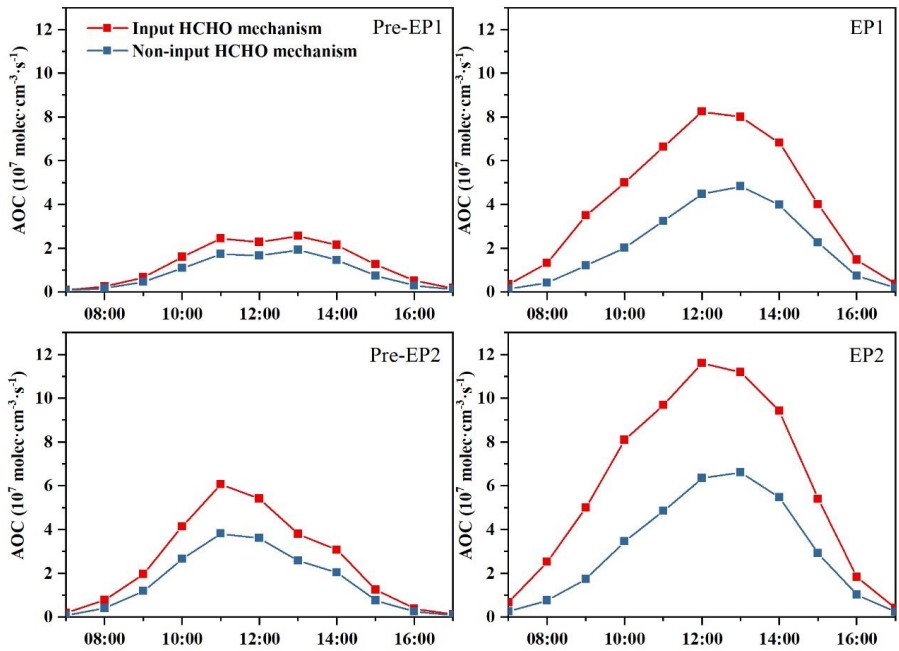

**Figure 8. Atmospheric oxidation capacity (AOC) calculated by the OBM with and without the HCHO mechanism.**

In addition, the maximum daily concentrations of OH, HO$_2$, and RO$_2$ exhibited a similar pattern to that of AOC in both the IH and NIH scenarios (Fig. 9). Therefore, the O$_3$ production rate during EP1 and EP2 was consistent with the maximum daily values of AOC, OH, HO$_2$, and RO$_2$. The differences in ROx levels between the IH and NIH model scenarios were also calculated (Fig. S11). In this study, disabling the HCHO mechanism led to decreased ROx concentrations, affecting the O$_3$ formation. These results highlight the significance of HCHO in the photochemical reactions occurring in the coastal city during the co-occurring O$_3$ and PM$_{2.5}$ pollution period.





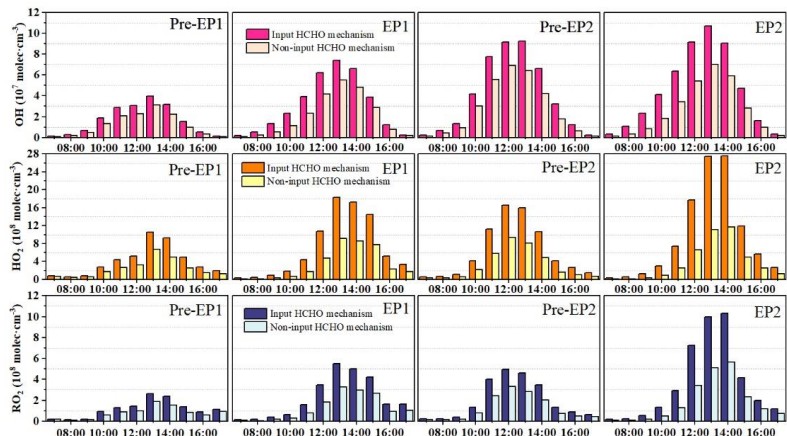

**Figure 9. OH, HO₂, and RO₂ concentrations modeled by the OBM with or without the HCHO mechanism**




## Conclusions

A wintertime co-occurring $O_3$ and $PM_{2.5}$ pollution event was selected to investigate
the synergistic effects between $PM_{2.5}$ and $O_3$ in a coastal city in southeast China. The
results domonstrated a positive correlation between $PM_{2.5}$ and MDA8 $O_3$
concentrations during the whole periods, indicating the enhancement of atmospheric
oxidation capacity (AOC) during cold seasons. The result of positive matrix
factorization (PMF) analysis suggested that the contribution of secondary formation to
$PM_{2.5}$ increased during the pollution events, implying that the elevated AOC promoted
the oxidation of $SO_2$, NOx, and VOCs, leading to the formation of secondary inorganic
and organic components. We also observed significant correlations ($R^2 = 0.415–0.477$)
between HCHO, Fe, Mn, and sulfate concentrations, suggesting the influence of
catalyzed oxidation in the coastal city. Through OBM analysis, we demonstrated that
high concentrations of precursors ($SO_2$ and HCHO), high RH, and moderately acidic
pH conditions enhanced the heterogeneous formation of hydroxymethanesulfonate
(HMS). Meanwhile, we verified that the input HCHO mechanism increased the
concentrations of ROx and net $O_3$ production rates. Moreover, the production rates of
$HO_2+NO$ and $RO_2+NO$ were enhanced, indicating that HCHO affected $O_3$ formation
by controlling the efficiencies of radical propagation. This study highlighted the
influence of the HCHO mechanism on co-occurring $O_3$ and $PM_{2.5}$ pollution in coastal
cities and was beneficial for improving air quality and protecting public health.



***Data Availability.*** The data set related to this work can be accessed via
https://doi.org/10.5281/zenodo.7799302 (Hong, 2023). The details are also available
upon request from the corresponding author (ywhong@iue.ac.cn).


***Authorship Contribution Statement.*** YW designed and wrote the manuscript. YL and
LD collected the data, and GJ and ZM contributed to the modeling analyses. KR, XT,
XK, and WY performed data analysis. SD, LK, RL, and GR contributed to revising the
manuscript. JS supported funding of observation and research.

***Competing interests.*** The authors declare that they have no conflict of interest.

***Acknowledgement.*** The authors gratefully acknowledge Yanting Chen and Zhiqian
Shao (Institute of Urban Environment, Chinese Academy of Sciences) for their
guidance and assistance during the observation, and Lingling Xu, Mengren Li, and
Xiaolong Fan (Institute of Urban Environment, Chinese Academy of Sciences) for the
discussion of this paper. This research was supported by the Xiamen Atmospheric
Environment Observation and Research Station of Fujian Province (Institute of Urban
Environment, Chinese Academy of Sciences).

***Financial support.*** This research was financially supported by the National Natural
Science Foundation of China (42277091, U22A20578), the foreign cooperation
project of Fujian Province (2020I0038), the Xiamen Youth Innovation Fund Project
(3502Z20206094), the FJIRSM&IUE Joint Research Fund (RHZX-2019-006), and
Center for Excellence in Regional Atmospheric Environment project (E0L1B20201).




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
