# Peer review of "Exploring the amplied role of HCHO during the wintertime ozone"

_EGUsphere, 2023_

## Author Comment (AC1)

Editor and Reviewer comments:

Response: We thank the editor and reviewers for good comments and suggestions. We have addressed each comment in the following point by point. In addition, we have adjusted our reference list according to the ACP guideline.

**RC1**: 'Comment on egusphere-2023-1242', Anonymous Referee #1, 14 Jul 2023

The manuscript by Hong et al. investigated the synergistic mechanisms between fine particulate matter ($PM_{2.5}$) and surface ozone ($O_3$) in a coastal city of southeast China. Especially, the authors explore the influence mechanism of HCHO on co-occurring $O_3$ and $PM_{2.5}$ pollution based on the observation-based model (OBM). They employed well established analytical techniques for identification and quantification of HCHO effects on hydroxymethanesulfonate (HMS) and $O_3$ formations. The obtained results are interesting and would be helpful for understanding the role of HCHO in photochemical pollution and secondary aerosol formation. I can recommend its publication in Atmospheric Chemistry and Physics (ACP) after addressing the comments below:

Response: Thank you very much for all the valuable comments and suggestions. We have addressed each comment in the following point by point and have revised the manuscript accordingly.

Line 37-38:the authors mentioned, "suggesting an increase in atmospheric oxidation capacity (AOC) during the cold seasons". Could you elaborate why or provide more evidence to support it. This is not clear.

Response: Thank you very much for your suggestions. We provided the supporting data for atmospheric oxidation capacity (AOC) during the cold seasons. Inter-annual averaged concentrations of $O_3$ and Ox in winter has been added into Fig S3.

So, the related sentences have been added in the revised manuscript, as follows:

The results of this study revealed the characteristics of positively correlated $PM_{2.5}$ and MDA8 $O_3$ concentrations, and an increase in atmospheric oxidation capacity (AOC) during the cold seasons.

Meanwhile, inter-annual averaged concentrations of $O_3$ and Ox in winter were shown in Fig S3 (b), suggesting an increase in atmospheric oxidation capacity (AOC) during the cold seasons.

[Figure]

Lines 79 to 90: Additional references about the effects of HCHO on HMS in the introduction are needed.

Response: Thank you for your good suggestions. We have added the details of the HMS in the revised manuscript, as follows:

Recent studies have shown that HCHO can react with hydrogen peroxide ($H_2O_2$) to produce hydroxymethyl hydroperoxide (HMHP), which rapidly oxidizes dissolved sulfur dioxide ($SO_2$, aq) to sulfate (Dovrou et al., 2022). Meanwhile, HCHO reacts with dissolved $SO_2$ (aq) to produce hydroxymethanesulfonate (HMS, $HOCH_2SO^-$), which, upon oxidation with the hydroxyl radical (OH), forms sulfate (Ma et al., 2020; Moch et al., 2020). Totally, atmospheric HCHO contributes to sulfate formation in $PM_{2.5}$ by producing $HO_2$ radicals and HMHP or HMS (Wu et al., 2023; Dovrou et al., 2022; Campbell et al., 2022). However, these studies highlight the necessity for more observation research to obtain evidence of the contributions of HCHO to HMS formation. HMS is an important organosulfur compound in the atmosphere, not only in cloud and fog but also in atmospheric aerosols (Munger et al., 1986; Dixon and Aasen, 1999). The misidentification of HMS as inorganic sulfate caused the overestimation of the observed particulate sulfate (Ma et al., 2020; Dovrou et al., 2022).

Munger, J. W., Tiller, C., and Hoffmann, M. R.: Identification of hydroxymethanesulfonate in fog water, Science, 231, 247–249, https://doi.org/10.1126/science.231.4735.247, 1986.

Dixon, R. W. and Aasen, H.: Measurement of hydroxymethanesulfonate in atmospheric aerosols, Atmos. Environ., 33, 2023–2029, https://doi.org/10.1016/s1352-2310(98)00416-6, 1999.

Lines 198 to 209: The methodology for HMS modeling needs to be presented in detail.

Response: Thank you for your kindly suggestions. We have added the descriptions in all the manuscript comprehensively.

To simulate the concentration of particulate HCHO and its role in the heterogeneous formation of hydroxymethanesulfonate (HMS), the observation-based zero dimensional multiphase chemical box model was used, of which the gas phase chemistry is described by he regional atmospheric chemistry mechanism version 2 (RACM2) and the aqueous phase part of the mechanism is represented by the chemical aqueous-phase radical mechanism version 3.0 (CAPRAM 3.0). The mass transfer processes between the gas and aqueous phases is also considered in current model according to Schwartz (1986). The Henry's law constant of HCHO was updated with a value of $0.31 \times 10^8$ M atm$^{-1}$, as estimated by Mitsuishi et al. (2018). Sensitivity analysis was conducted to evaluate the uncertainties introduced by the Henry's law constant, details can be found in SI. For major production and loss paths of HMS, dissolved HCHO reacts with sulfite and bisulfite to form HMS (Eq (3-4)), which can be further oxidized by aqueous OH radicals (Eq (5)), details about the HMS mechanisms and the corresponding reaction kinetics are documented in CAPRAM website (https://capram.tropos.de/).

$$HCHO_{(aq)} + HSO_3^- = HOCH_2SO_3^- \tag{3}$$

$$HCHO_{(aq)} + SO_3^{2-} + H_2O = HOCH_2SO_3^- + OH^- \tag{4}$$

$$HOCH_2SO_3^- + OH \cdot = 2SO_4^{2-} + HCHO_{(aq)} + 3H^+ \tag{5}$$

For detailed modeling steps of HMS, firstly, the observation data of gaseous NO, NO$_2$, O$_3$, SO$_2$, CO, NMHCs, HCHO and other ten carbonyls, particulate phase NO$_3^-$, NH$_4^+$, and Cl$^-$, along with meteorological parameters were averaged or interpolated into a 1-h time resolution and classified into model recognized groups as model inputs, while the measured SO$_4^{2-}$ was used as the initial conditions. Liquid water content (LWC) and aqueous H$^+$ concentrations, calculated using the ISORROPIA-II model (Hong et al., 2022), were also used as model inputs. Then, model calculations were conducted using the commercial FACSIMILE software, the modeling period was from February 26 to March 16, 2022 and each day was regarded as an independent simulation case. The model was constrained every hour by the input observation data for integral calculation. For each case, The integration ran three times in series to steady the unconstrained species (e.g., radicals) , which was initiated at 00:00 local time (LT), and had a step of 1 h and a duration of 24 h. Finally, the modeled HMS concentrations of the third run were outputted with a 1-hour time resolution for further analysis.

Line 348-355: The manuscript focused on the HCHO mechanisms, so it is suggested to discuss more about the effects of HCHO on HMS in $PM_{2.5}$.

Response: Thank you for your good suggestions. Indeed, as the reviewer mentioned, the effects of HCHO on HMS in $PM_{2.5}$ were emphasized in this study. We have added the descriptions in all the manuscript comprehensively.

However, the molar ratio of HMS to sulfate were very low, suggesting the limited contributions of HMS concentrations to inorganic sulfate concentrations. Potential roles of HCHO in the HMS formation in coastal cities of southeast China was differed from those in the megacities of China. Previous studies found that HCHO reacts with dissolved $SO_2$ (aq) to produce hydroxymethanesulfonate (HMS), which, upon oxidation with the hydroxyl radical (OH), forms sulfate (Ma et al., 2020; Moch et al., 2020). Ma et al. (2020) reported that heterogeneous formation of HMS accounted for 15% of OM, and resulted in 36% overestimates of sulfate during the winter haze in Beijing.

Line 313-314:The authors state " As shown in Fig. 3(b) and (c), a good correlation was found between $SO_4^{2-}$ and Fe and Mn.". The use of correlations is indeed a helpful tool to explore some specific trends; however, such data processing techniques are not sufficient to reach a definite conclusion on that the TMI-catalyzed oxidation contributed to the formation of $SO_4^{2-}$.

Response: Thank you for your good comments. The reviewer raised a good point that the TMI-catalyzed oxidation contributed to the formation of $SO_4^{2-}$. As shown in Figure 1, under low aerosol pH, the catalytic reaction of transition metal ions (TMIs), e.g., Fe(III) and Mn(II) plays an important role in the oxidation of S(IV). In this study, low aerosol pH ranged from 2 to 4 was observed, indicating the potential influence of TMI-catalyzed oxidation.
Previous studies found that Mn(III) was the intermediate product that could oxidize S(IV) (Wang et al., 2021). Meanwhile, $SO_3^-$ can react with dissolved $O_2$ to generate $SO_5^-$ radicals, which can oxidize Mn(II) to regenerate Mn(III) (Seinfeld and Pandis, 2016). Therefore, the Mn catalytic redox reaction could continue to produce sulfate while consuming only oxygen and $SO_2$, meaning that this reaction would occur even in the low concentration of Mn (Wang et al., 2021). The sulfate production rate was affected by many factors, including the aerosol surface, the aerosol acidity and ionic strength, and environmental conditions (temperature and relative humidity), along with various mixing ratios of $SO_2$, $NO_2$ and $NH_3$ (Zhang et al., 2020; Hung et al., 2018; Lee et al., 2019). Some findings demonstrated that there may be a massive gap in the reaction rate between air–liquid interface and bulk solution, and the mechanism of air–liquid interface rate enhancement effect may differ greatly in a distinct reaction system (Yan et al., 2017; Zhang et al., 2020; Lee et al., 2019). For aqueous phase oxidation to form sulfate processes, such as oxidization by $NO_2,O_3,H_2O_2$, are not only limited by the availability of oxidants in the atmosphere and the solubility of $SO_2$, these oxidants would also be consumed at the same order of magnitude as the

formation of sulfate. In the future, it is vital to further evaluate the interaction of sulfate formation and Fe/Mn, and to elucidate the main pathway of the S(IV) oxidation in the coastal areas.

[Figure]

Figure1 The formation of $SO_4^{2-}$ in fine particulate matter under different aerosol acidity conditions

Seinfeld, J. H. & Pandis, S. N. Atmospheric Chemistry and Physics: From Air Pollution to Climate Change (John Wiley & Sons, 2016).

Yan, X., Cheng, H. & Zare, R. N. Two-phase reactions in microdroplets without the use of phase-transfer catalysts. Angew. Chem. 129, 3616–3619 (2017).

Zhang, Y., Apsokardu, M. J., Kerecman, D. E., Achtenhagen, M. & Johnston, M. V. Reaction kinetics of organic aerosol studied by droplet assisted ionization: enhanced reactivity in droplets relative to bulk solution. J. Am. Soc. Mass Spectrom. 32,46–54 (2020).

Hung, H. M., Hsu, M. N. & Hoffmann, M. R. Quantification of $SO_2$ oxidation on interfacial surfaces of acidic micro-droplets: implication for ambient sulfate formation. Environ. Sci. Technol. 52, 9079–9086 (2018).

Lee, J. K. et al. Spontaneous generation of hydrogen peroxide from aqueous microdroplets. Proc. Natl Acad. Sci. USA 116, 19294–19298 (2019).

Y. Cheng, G. Zheng, C. Wei, Q. Mu, B. Zheng, Z. Wang, et al. Reactive nitrogen chemistry in aerosol water as a source of sulfate during haze events in China.[J].Science advances.2016,(12).e1601530.

Wang, W., Liu, M., Wang, T., et al., 2021. Sulfate formation is dominated by manganesecatalyzed oxidation of $SO_2$ on aerosol surfaces during haze events. Nat. Commun. 12, 1993. https://doi.org/10.1038/s41467-021-22091-6.

We discussed the effects of metal oxidation on the sulfate production in the revised manuscript, as follows:

Under low aerosol pH conditions, the catalytic reaction of TMIs plays an important role in the oxidation of S(IV). In this study, low aerosol pH ranged from 2 to 4 was observed, indicating the potential influence of TMI-catalyzed oxidation. In the future,

it is vital to further evaluate the interaction of sulfate formation and Fe/Mn, and to elucidate the main pathway of the S(IV) oxidation in the coastal areas.

---

## Author Comment (AC2)

RC2: ['Comment on egusphere-2023-1242'](), Anonymous Referee #2, 25 Jul 2023

The authors examined the role of HCHO in the formation of PM2.5 and ozone using integrated measurements in a coastal city of China. They conducted box model runs to illustrate the chemical role of HCHO in PM2.5 formation, ozone, as well as atmospheric oxidation capacity. Overall, I believe the topic of this study fits well with the scope of ACP. The manuscript is also easy-reading. I am happy to see its publication in due course. However, in current version, I think more discussion on HCHO should be added and its role in $PM_{2.5}$ needs to be further justified. Please find my comments below:

Response: Thank you very much for all the valuable comments and suggestions. We have addressed each comment in the following point by point and have revised the manuscript accordingly.

The authors highlight the important role of HCHO in $PM_{2.5}$, but HMS concentrations are very limited and I am also not convinced that HCHO plays very important role in inorganic sulfate formation. Please justify your argument.

Response: Thank you for your good suggestions. We are sorry about some unclear expressions in the original manuscript. Indeed, as the reviewer mentioned, HMS concentrations are very limited and their particulate sulfur molar percentage were relatively low. We just try to present the important role of HCHO on the formations of HMS (not for the formation of $PM_{2.5}$) and ROx radicals during the wintertime co-occurring ozone and $PM_{2.5}$ pollution. According to the reviewer's opinion, we revised the title of the manuscript.

"Exploring the amplied role of HCHO in the formation of HMS and $O_3$ during the co-occurring $PM_{2.5}$ and $O_3$ pollution in a coastal city of southeast China"

In addition, we have revised the related descriptions in all the manuscript comprehensively, as follows:

However, the molar ratio of HMS to sulfate were very low (Figure 4), suggesting the limited contributions of HMS concentrations to inorganic sulfate concentrations. Potential roles of HCHO in the HMS formation in coastal city of southeast China was differed from those in the megacities of China. Previous studies found that HCHO reacts with dissolved $SO_2$ (aq) to produce hydroxymethanesulfonate (HMS), which, upon oxidation with the hydroxyl radical (OH), forms sulfate (Ma et al., 2020; Moch et al., 2020). Ma et al. (2020) reported that heterogeneous formation of HMS accounted for 15% of OM, and resulted in 36% overestimates of sulfate during the winter haze in Beijing. The HMS concentration and the molar ratio of HMS to sulfate increased with the deterioration of winter haze, as well as from winter 2015 to winter 2016 with the growth in HCHO concentration. Meanwhile, Moch et al. (2018) and Song et al. (2019) reported the potential contribution of hydroxymethanesulfonate (HMS) to particulate sulfur during winter haze in Beijing.

The authors ran box model with and without the HCHO mechanism. But how was this conducted? E.g., disable the HCHO photolysis? The method should be well explained.

Response: Thank you for your comments.

To furtherly quantify the varieties in ROx chemistry and $O_3$ formation in response to HCHO chemistry, two parallel scenarios were conducted through the F0AM model. One scenario was run with all MCM mechanism defined as Input HCHO mechanism, and the other was run with the HCHO mechanism disabled in MCM mechanism defined as Non-input HCHO mechanism. Considering the numerous chemical formation reactions of HCHO, our study mainly disabled the HCHO loss pathways of HCHO photolysis, HCHO + OH, and HCHO + $NO_3$, which makes HCHO become a stable secondary pollutant that will not continue to react, thus the HCHO loss pathways were disabled in Non-input HCHO mechanism scenario. The differences in ROx levels and production pathways in the two model scenarios were analyzed to investigate the chain effect of HCHO on ROx cycling and $O_3$ formation. More details could be found in our previous studies (Liu et al., 2023).

The related sentences have been added into the revised manuscript, as follows:

To furtherly quantify the effects of HCHO on ROx chemistry and $O_3$ formation, we disabled the loss pathways of HCHO photolysis, HCHO + OH, and HCHO + $NO_3$ in MCM mechanism, defined as Non-input HCHO mechanism (Liu et al., 2023). Meanwhile, the other scenario was run with all MCM mechanisms, defined as Input HCHO mechanism.

Liu, T., Lin, Y., Chen, J.*, Chen, G., Yang, C., Xu, L., Li, M., Fan, X., Zhang, F., and Hong, Y.* Pollution mechanisms and photochemical effects of atmospheric HCHO in a coastal city of southeast China. Science of the Total Environment. 2023, 859:160210

I am very interested in where does HCHO come from? Secondary vs primary origin? How is the HCHO level compared with other studies?

Response: Thank you for your good suggestions. However, we didn't quantify their secondary and primary sources under different pollution stages using the PMF method, due to the limited data. By the way, we have another preparing manuscript that focused on the occurrence, seasonal variations of HCHO in the monitoring site, and would comprehensively quantify their secondary and primary sources based on the PMF method or the photochemical age-based parameterization method. Preliminary results showed that the contribution of secondary formation was the largest, followed by vehicle exhaust, biogenic and industrial emissions, and solvent usage. In this study, for the co-occurring ozone and $PM_{2.5}$ pollution cases, we just try

to emphasize the important role of HCHO on the formations of HMS and ROx radicals.

During the monitoring periods, the concentrations of measured HCHO ranged from 0.68 ppbv and 3.59 ppbv (Tabe S1). According to our previous studies (Liu et al., 2023), the average levels of the measured HCHO in spring and autumn in Xiamen were $2.9 \pm 0.3$ ppbv and $3.2 \pm 1.4$ ppbv, respectively. Totally, the HCHO level in Xiamen was lower than that in megacities (Table S2), such as Beijing (summer: 11.39±5.58 ppbv), Hongkong (summer: 8.07±1.94 ppbv), and Guangzhou (summer: 6.69±1.98 ppbv), while was comparable to the coastal cities, including Shenzhen (spring: 3.4±1.6 ppbv), Yantai (summer: 3.90±1.12 ppbv), and Shanghai (summer:3.31±1.43 ppbv). More details have been added in the revised manuscript, as shown in Table S2. Also, we have added the related descriptions in the revised manuscript.

Table S2 Comparisons of atmospheric HCHO in China

| Location | Type | Seasons | Mean(±SD) (ppbv) | Range (ppbv) | Reference |
|---|---|---|---|---|---|
| Xiamen, China | Urban | Spring | 2.9 ± 0.3 | 0.25-8.34 | [12] |
| | Urban | Autumn | 3.2 ± 1.4 | 0.38-7.56 | [12] |
| | Urban | Winter | 1.95±1.06 | 0.23-6.22 | This study |
| Shenzhen, China | Urban | Spring | 3.4±1.6 | N.A.[a] | [2] |
| Hong Kong, China | Urban | Spring (2012) | 3.02±0.91 | N.A. | [6] |
| Shanghai, China | Suburban | Spring (2018) | 6.7±3.6 | N.A.-20.9 | [4] |
| Shanghai, China | Suburban | Spring (2018) | 5.01±3.80 | N.A.-18.69 | [10] |
| Beijing, China | Urban | Summer (2013) | 11.39±5.58 | N.A. | [5] |
| Shanghai, China | Urban | Summer (2018) | 3.31±1.43 | N.A. | [8] |
| Shenzhen, China | Urban | Summer | 5.0±4.4 | N.A. | [2] |
| Hong Kong, China | Urban | Summer (2011) | 8.07±1.94 | N.A. | [6] |
| Guangzhou, China | Urban | Summer (2010) | 6.69±1.98 | N.A. | [3] |
| Yantai, China | Urban | Summer | 3.90±1.12 | N.A. | [3] |
| Beijing, China | Suburban | Summer | 11.17±5.32 | 3.14-35.08 | [9] |
| Shanghai, China | Suburban | Summer | 2.2±1.8 | N.A.-9.4 | [7] |
| Wuhan, China | Suburban | Summer | 2.1±0.2 | 0.6-4.1 | [1] |
| Hong Kong, China | Urban | Autumn (2011) | 2.96±0.70 | N.A. | [6] |
| Guangdong,China | Urban | Autumn | 4.12±1.02 | 2.56-7.31 | [11] |
| Beijing, China | Urban | Winter (2013) | 7.39±5.26 | N.A. | [5] |
| Shenzhen, China | Urban | Winter | 4.2±2.2 | N.A. | [2] |
| Hong Kong, China | Urban | Winter (2012) | 2.70±1.20 | N.A. | [6] |
| Guangzhou, China | Urban | Winter | 3.35±1.38 | N.A. | [3] |

Note: (a) N.A. means no relevant data available.

[1]Zeng P, Lyu X, Guo H, et al. Spatial variation of sources and photochemistry of formaldehyde in Wuhan, Central China[J]. Atmospheric Environment, 2019, 214: 116826.

[2]Wang C, Huang X, Han Y, et al. Sources and Potential Photochemical Roles of Formaldehyde in an Urban Atmosphere in South China[J]. Journal of Geophysical Research: Atmospheres, 2017, 122(21): 11,934-11,947.

[3]Ho K F, Ho S S H, Huang R J, et al. Spatiotemporal distribution of carbonyl compounds in China[J]. Environmental Pollution, 2015, 197: 316-324.

[4]Zhang K. Formation mechanism of HCHO pollution in the suburban Yangtze River Delta region, China: A box model study and policy implementations[J]. Atmospheric Environment, 2021, 267: 118755.

[5]Rao Z, Chen Z, Liang H, et al. Carbonyl compounds over urban Beijing: Concentrations on haze and non-haze days and effects on radical chemistry[J]. Atmospheric Environment, 2016, 124: 207-216.

[6]Cheng Y, Lee S C, Huang Y, et al. Diurnal and seasonal trends of carbonyl compounds in roadside, urban, and suburban environment of Hong Kong[J]. Atmospheric Environment, 2014, 89: 43-51.

[7]Wu Y, Huo J, Yang G, et al. Measurement report: Production and loss of atmospheric formaldehyde at a suburban site of Shanghai in summertime[J]. Atmospheric Chemistry and Physics, 2022, 23: 2997-3014.

[8]Guo Y, Wang S, Zhu J, et al. Atmospheric formaldehyde, glyoxal and their relations to ozone pollution under low- and high-NOx regimes in summertime Shanghai, China[J]. Atmospheric Research, 2021, 258: 105635.

[9]Yang X, Xue L, Wang T, et al. Observations and Explicit Modeling of Summertime Carbonyl Formation in Beijing: Identification of Key Precursor Species and Their Impact on Atmospheric Oxidation Chemistry[J]. Journal of Geophysical Research: Atmospheres, 2018, 123(2): 1426-1440.

[10]Zhang K, Huang L, Li Q, et al. Explicit modeling of isoprene chemical processing in polluted air masses in suburban areas of the Yangtze River Delta region: radical cycling and formation of ozone and formaldehyde[J]. Atmospheric Chemistry and Physics, 2021, 21(8): 5905-5917

[11]Shen H, Liu Y, Zhao M, et al. Significance of carbonyl compounds to photochemical ozone formation in a coastal city (Shantou) in eastern China[J]. Science of The Total Environment, 2021, 764: 144031.

[12]Liu, T., Lin, Y., Chen, J.*, Chen, G., Yang, C., Xu, L., Li, M., Fan, X., Zhang, F., and Hong, Y.* Pollution mechanisms and photochemical effects of atmospheric HCHO in a coastal city of southeast China. Science of the Total Environment. 2023, 859:160210

L219-224: is this backward trajectory analysis necessary?

Response: Thank you for your kindly suggestions. Originally, backward trajectory analysis was used for sources apportionment of PM$_{2.5}$ under different pollution stages. So, we removed it to the SI.

L244: I suggest to replace "might be" with "are".

Response: Corrected.

In Fig.1, it looks halogenated VOC contributed greatly to TVOCs. What are they coming from? Do you include HCHO concentrations in your TVOCs calculation?

Response: Thank you for your good comments. In the coastal cities of southeastern China, halogenated VOC is one of important VOC species, which originated from industrial emissions and solvent usage, according to our previous studies (Chen et al., 2022; Ji et al., 2022; Liu et al., 2022). In this study, during the monitoring period, backward trajectories showed air mass transport from the northeast, which brought pollutants from Quanzhou city, an industrial city adjacent to Xiamen. Similar to aromatics (2131±1236 pptv), the concentrations of halocarbons (1951±572 pptv) was higher than alkenes (1205±464 pptv) and acetylene (674±290 pptv), according to our previous studies (Liu et al., 2022). So, we have added the related descriptions in the revised manuscript.

In addition, we didn't include HCHO in the TVOCs calculation.

Chen, G., Liu, T., Ji, X., Xu, K., Hong, Y*., Xu, L., Li, M., Fan, X., Chen, Y., Yang, C., Lin, Z., Huang, W., and Chen, J.: Source Apportionment of VOCs and O3 Production Sensitivity at Coastal and Inland Sites of Southeast China, Aerosol Air Qual. Res., 22, 220289, 10.4209/aaqr.220289, 2022.

Ji, X., Xu, K., Liao, D., Chen, G., Liu, T., Hong, Y. *, Dong, S., Choi, S.-D., and Chen, J.*: Spatial-temporal Characteristics and Source Apportionment of Ambient VOCs in Southeast Mountain Area of China, Aerosol Air Qual. Res., 22, 220016, 2022.

Liu T.#, Hong, Y.#, Li, M., Xu, L., Chen, J.*, Bian Y., Yang C., Dan Y., Zhang Y., Xue L., Zhao M., Huang Z., Wang H., Atmospheric oxidation capacity and ozone pollution mechanism in a coastal city of Southeast China: Analysis of a typical photochemical episode by Observation-Based Model. Atmos. Chem. Phys. 22, 2173-2190, 2022

L300-301: "under different periods" refers to EP1 and EP2?

Response: Yes. We have clarified this sentence, as follows:

However, there was no significant difference in the existing form of SNA in $PM_{2.5}$ during EP1 and EP2.

L334: change "HCOH" to "HCHO"

Response: Corrected.

---

## Author Comment (AC3)

**RC3**: 'Comment on egusphere-2023-1242', Anonymous Referee #3, 03 Aug 2023

Hong et al examine two pollution episodes in Xiamen, China characterized by elevated PM and ozone. They look at the impact of formaldehyde on both, using a box model to demonstrate the role of HCHO on hydroxymethanesulfonate formation in PM2.5 and the importance of HCHO on ozone production.  While the authors do a fine job of presenting their results, the main weakness of this paper is the strong dependence on a box model without any attempts at validation or discussion of the potential limitations/uncertainties.  Before publication, concerns related to this issue, outlined in more detail below, and other minor points need to be addressed.

Response: Thank you very much for all the valuable comments and good suggestions. We have addressed each comment in the following point by point and have revised the manuscript accordingly.

Indeed, we didn't present the details of the box model analysis, and have also limited discussion of its potential limitations/uncertainties. Here, we make enough supplement for the related materials of the box model, according to your good suggestions.

**The Observation-based model (OBM)**

A chemical box model, as one of the important methods for analyzing atmospheric chemical processes, was run based on the platform of the Framework for 0-Dimensional Atmospheric Modeling (F0AM), which has broad application potential in deeply exploring atmospheric observation data and comprehensively understanding the regional atmospheric pollution. About the chemical mechanism, the F0AM incorporating the latest chemical mechanism version of MCM-v3.3.1 (MCM, http://mcm.leeds.ac.uk/MCM/, last access: 13 May 2022) was applied to simulate the detailed photochemical processes and quantify the reaction rates of HCHO mechanism, and the MCM mechanism introduced 142 VOCs and about 20,000 chemical reactions.

About the uncertainties of the model simulation results, the index of agreement (IOA) was used to judge the reliability of the model simulation results, as follows:

$$IOA = 1 - \frac{\sum_{i=1}^{n}(O_i - S_i)^2}{\sum_{i=1}^{n}(|O_i - \bar{O}| - |S_i - \bar{O}|)^2}$$

where $S_i$ is simulated value, $O_i$ represents observed value, $\bar{O}$ is the average observed values, and n is the sample number. The IOA range is 0-1, and the higher the IOA value is, the better agreement between simulated and observed values is. In this study, the simulation results (the IOA is approximately 0.80) are reasonable, and the performance of the OBM-MCM model was acceptable.

**Uncertainty evaluation of aqueous HCHO analysis:**

Since aqueous HCHO was not available during the observation, which was the key chemical components influencing the subsequent HMS modeling, we established the aqueous HCHO concentrations by the mass transfer processed between the gas- and particle- phase. Among this processes, the uncertainties were introduced somehow by the Henry's law constant adopted in the model. We conducted a sensitivity test with 2 folds of current used Henry's law constant of HCHO, with a value of $6 \times 10^7$ M atm$^{-1}$. As shown in Fig S0, the modeled aqueous HCHO as well as HMS concentrations increased with the increase of Henry's law constant in the sensitivity case, with increase of 0.01 and 0.13 µg m$^{-3}$ for aqueous HCHO and HMS, respectively. It is quite reasonable considering the increased solubility of HCHO. On the other hand, the modeled HCHO and HMS were still exhibited higher concentrations during the pollution episodes (EP1 and EP2), of which higher precursors and favorable aerosol properties enhanced the heterogeneous processes. Therefore, the impacts of HCHO Henry's law constant approximations on the conclusions are supposed to be minor.

[Figure]

**Fig S0. Time series of the modeled aqueous HCHO and HMS concentrations with the model used Henry's law constant of HCHO (referred as base case, the blue line) and 2 folds of the model used Henry's law constant of HCHO (referred as sensitivity case, the red line), respectively.**

Line 85: I am unfamiliar with HMS, as I imagine, will be many of the readers of this paper. More background should be given as to the importance and relevance of this species in the introduction.

Response: Thank you for your good suggestions. We have added the details of the HMS in the revised manuscript, as follows:

HMS is an important organosulfur compound in the atmosphere, not only in cloud and fog but also in atmospheric aerosols (Munger et al., 1986; Dixon and Aasen, 1999). The misidentification of HMS as inorganic sulfate caused the overestimation of the observed particulate sulfate (Ma et al., 2020; Dovrou et al., 2022).

Recent studies have shown that HCHO can react with hydrogen peroxide ($H_2O_2$) to produce hydroxymethyl hydroperoxide (HMHP), which rapidly oxidizes dissolved sulfur dioxide ($SO_2$, aq) to sulfate (Dovrou et al., 2022). Meanwhile, HCHO reacts

with dissolved SO$_2$ (aq) to produce hydroxymethanesulfonate (HMS, HOCH2SO−), which, upon oxidation with the hydroxyl radical (OH), forms sulfate (Ma et al., 2020; Moch et al., 2018, 2020).

Dixon, R. W. and Aasen, H.: Measurement of hydroxymethanesulfonate in atmospheric aerosols, Atmos. Environ., 33, 2023–2029, https://doi.org/10.1016/s1352-2310(98)00416-6, 1999.

Munger, J. W., Tiller, C., and Hoffmann, M. R.: Identification of hydroxymethanesulfonate in fog water, Science, 231, 247–249, https://doi.org/10.1126/science.231.4735.247, 1986.

Line 106: You introduce the term "observation-based model" analysis here like it is a standard term. You need to explain here that this is the name of the modeling framework that you are using for this study.

Response: Thank you for your kindly comments. We have added the details of the OBM model in SI.

A chemical box model, as one of the important methods for analyzing atmospheric chemical processes, was run based on the platform of the Framework for 0-Dimensional Atmospheric Modeling (F0AM), which has broad application potential in deeply exploring atmospheric observation data and comprehensively understanding the regional atmospheric pollution. About the chemical mechanism, the F0AM incorporating the latest chemical mechanism version of MCM-v3.3.1 (MCM, http://mcm.leeds.ac.uk/MCM/, last access: 13 May 2022) was applied to simulate the detailed photochemical processes and quantify the reaction rates of HCHO mechanism, and the MCM mechanism introduced 142 VOCs and about 20,000 chemical reactions (Jenkin et al., 2003).

Jenkin, M.E., Saunders, S.M., Wagner, V., Pilling, M.J., 2003. Protocol for the development of the Master Chemical Mechanism, MCM v3 (Part B): tropospheric degradation of aromatic volatile organic compounds. Atmos. Chem. Phys. 3, 181–193.

Line 117: Since your study is focused on Feb – Mar, isn't the average RH and T for that time period more relevant than the annual average.

Response: Thank you for your good suggestions. We have revised the sentence, as follows:

It is situated in a subtropical monsoon climate, with an average temperature of 18.5°C and a relative humidity of 63.3% during the wintertime observation.

Line 132: Information about the uncertainties of your observations is needed, particularly for formaldehyde.

Response: Thank you for your comments. We have added the details of measured air pollutants in TableS3, as follows:

Table S3. The detection limits, time resolutions and measured uncertainties of air pollutants

| Species | Measurement Techniques | Uncertainties | Detection limits | Time resolution |
|---|---|---|---|---|
| HCHO | FMS-100, Focused Photonics Inc., Hangzhou, China | ≤5% | 50 pptv | 1 s |
| PAN | PANs-1000, Focused Photonics Inc., Hangzhou, China | ±10% | 50 pptv | 5 min |
| $O_3$ | Model 49i, Thermo Fischer Scientific, USA | ±5% | 1 ppbv | 1 h |
| NOx | Model 42i, Thermo Fischer Scientific, USA | ±10% | 0.5 ppbv | 1 h |
| CO | Model 48i, Thermo Fischer Scientific, USA | ±5% | 40 ppbv | 1 h |
| $SO_2$ | Model 43i, Thermo Fischer Scientific, USA | ±10% | 0.5 ppbv | 1 h |
| VOCs | GC-FID/MS, TH-300B, Wuhan, China | ±10% | 20-300 pptv | 1 h |
| HONO | MARGA, ADI 2080, Applikon Analytical B.V., the Netherlands | ±20% | 50 pptv | 1 h |

Line 155: 23:00 local time?

Response: Thanks. We have replaced it with 11:00 pm.

Line 167: What's the resolution of the ERA reanalysis and how dependent are your results on the accuracy of the PBL height? I would imagine deposition in your model could be quite sensitive to the accuracy of this term.

Response: Thank you for your good suggestions. The resolution of ERA-5 reanalysis is 0.25°×0.25°.However, the BLH data was only used to explain the diffusion conditions during different pollution levels of $PM_{2.5}$ and $O_3$. As you mentioned that, there exists potential uncertainties if they are considered as constraints data. Therefore, we didn't input them into the OBM model. The related sentences have been revised as follows:

According to our previous studies, the model incorporates the physical process of deposition within the boundary layer height (BLH), which varies from 300 m during nighttime to 1500 m during the daytime in winter (Li et al., 2018;Liu et al., 2022).

Line 195: I'm confused as to why you are using boundary layer heights from autumn when your study is based on Feb. – Mar.

Response: Thank you for your comments. Corrected.

Section 2.4: I agree with the other reviewer that more details about this model are needed. I assume this is a box model, although you never say that explicitly. How do you handle NO/NO2 constraints in this model? Are they both constrained to observations and held constant? Is NOx constrained as a family like in other box models (e.g. F0AM, DSMACC)? The handling of NOx will have considerable impact on your discussion of ozone production so more details are required here.

Response: Thank you for your good suggestions and comments. We have presented the details of the box model analysis, and made correspondingly the supplement for the related materials of the box model in the revised manuscript and SI.

As the reviewer mentioned, the NOx will have considerable impacts on the production of $O_3$. We used the observation data of NO and $NO_2$ in the real environment to handle NO/$NO_2$ constraints in this model (Line 222-225), according to our previous studies (Liu et al., 2022a,b). So, in this study, NOx constrained as a family is not applied for the box models.

Liu, T., Hong, Y., Li, M., Xu, L., Chen, J., Bian, Y., Yang, C., Dan, Y., Zhang, Y., Xue, L., Zhao, M., Huang, Z., and Wang, H.: Atmospheric oxidation capacity and ozone pollution mechanism in a coastal city of southeastern China: analysis of a typical photochemical episode by an observation-based model, Atmos. Chem. Phys., 22, 2173-2190, 10.5194/acp-22-2173-2022, 2022a.

Liu, T., Lin, Y., Chen, J., Chen, G., Yang, C., Xu, L., Li, M., Fan, X., Zhang, F., and Hong, Y.: Pollution mechanisms and photochemical effects of atmospheric HCHO in a coastal city of southeast China, Sci. Total Environ., 160210, https://doi.org/10.1016/j.scitotenv.2022.160210, 2022b.

Line 212: To what time-scale are you interpolating?

Response: Thanks. The observation data with 1h time resolution were interpolated to constrain the OBM model.

Line 221: What meteorology are you using and at what resolution?

Response: Thank you for your comments. Meteorological data used here were obtained from the Global Data Assimilation System (GDAS) with a 1°×1°spatial resolution and 3-h temporal resolution.

Line 223: You need to include a citation for cluster analysis.

Response: Thanks. Corrected.

Line 302: I might have missed it, but I don't think you ever defined what SOR and NOR mean.

Response: Thanks for your suggestions. Corrected.

Line 324: More information about the HCHO + H2O2 reaction is needed here. How easily does this reaction happen in ambient air, ie, is there actually significant production of hydroxymethyl hydroperoxide from this reaction? Is this a gas phase or aqueous phase reaction?

Response: Thanks for your suggestions. We have added the details of HCHO + $H_2O_2$ reaction in SI, as follow:

In the atmosphere, gaseous HCHO with high Henry's law constant can partition into aerosol or cloud/fog water, then the aqueous HCHO can reacts with $H_2O_2$ to form HMHP, with an upper bound forward rate constant of 100 (±35) $M^{-1}$ $s^{-1}$ and reverse rate constant of 0.6 (±0.2) $s^{-1}$(Dovrou et al.,2022). In the gas phase, HMHP is mainly formed by the hydration of $CH_2OO$ Criegee radicals and then can partition into aerosol water, which have the potential to oxidize $SO_2$(aq) to form sulfate. From the perspective of contributions of HMHP paths to sulfate formation, the researcher revealed that the HCHO-catalysis path (HCHO + $H_2O_2$) under fast equilibrium is more significant than HMHP-direct path (hydration of CH2OO in the gas-phase) to global sulfate, so HCHO-catalysis path is significant for HMHP production. So, in the future, we will carry out the observation of $H_2O_2$ for further evaluate the effects of HCHO + $H_2O_2$ reaction on the HMHP production.

Dovrou, E., Bates, K. H., Moch, J. M., Mickley, L. J., Jacob, D. J., and Keutsch, F. N.: Catalytic role of formaldehyde in particulate matter formation, P. Natl. Acad. Sci. USA, 119, e2113265119, 10.1073/pnas.2113265119, 2022.

Section 3.4: Here is where more investigation is needed. While I realize that you don't have aqueous phase measurements of HCHO, you need to present some form of evaluation of your model or at the very least an uncertainty analysis to put the accuracy of your results in context. Are there previous studies you can cite that use this model that can reproduce observations of any of the species you are modeling here? How do

measurement uncertainties affect your results?   If you perturb your input HCHO by its uncertainty, how does that change your aqueous phase HCHO, for example?   How is gamma determined in your model?   Is it just a set value or is it parameterized in the model somehow?   How does uncertainty in gamma affect your results?   What's the uncertainty in the Henry's law constant for HCHO?   Etc.

Response: Thank you for your good comments and suggestions. We have added the sentences in the revised manuscript.

Exactly, the lack of aqueous phase measurements of HCHO would introduce uncertainties somehow to the model simulation and the corresponding results. Before we initiated the aqueous HCHO and HMS modeling, we carefully considered the establishment and the reproduce of aqueous HCHO, which was the key steps for further HMS modeling. Since the Henry's law constant is the key parameter for HCHO partitioning, we noted that the theoretical value provided by the CAPRAM would not suitable for current aerosol particle modeling and might resulted the underestimate of aqueous HCHO, as many recent studies pointed out that the field-derived effective Henry's law constants of HCHO are almost several orders of magnitude ($10^4$ - $10^6$) higher than the theoretical values, also HCHO in ambient particle phase have been found several orders of magnitude higher than that predicted by the traditional theories. Therefore, we concluded the field-derived effective Henry's law constants of HCHO from the recent published studies, which mainly in the order of $10^7$ M atm$^{-1}$ at ambient temperatures (~296 K) during the observation period. We finally selected a middle value provided by Mitsuishi et al. (2018) with a value $3.1 \times 10^7$ M atm$^{-1}$. We performed the sensitivity analysis to evaluate the uncertainties introduced by the Henry's law constant of HCHO, details were added into the supplement materials. Here in brief, we used 2 folds of the HCHO Henry's law constant as the sensitivity case and conducted the simulation of the aqueous HCHO as well as HMS concentrations. We found that aqueous HCHO and HMS exhibited slight higher concentrations due to the increased solubility of HCHO, with increases of 0.01 and 0.13 µg m$^{-3}$ for HCHO and HMS, respectively. However, the modeled HCHO and HMS were still exhibited higher concentrations during the pollution episodes (EP1 and EP2), which still able to support our major conclusions. On the other hand, we compared our modeling aqueous HCHO concentrations with previous observations. As shown in the following tables, our modeling aqueous HCHO concentrations are comparable and in the same order of magnitude with previous observations. Therefore, uncertainties due to the lack of aqueous HCHO measurement and the HCHO Henry's law constant approximations would take minor impacts on our major conclusions.

Table. Comparison of modeled aqueous HCHO concentrations with previous observations

| Aqueous HCHO concentration (ng m$^{-3}$) | References |
|:---:|:---:|
| 18 | this study |
| 28 | Andrade et al., 1995 |
| 18 | Odabasi et al., 2005 |
| 40 | Klippel et al., 1980 |
| 23 | Shen et al., 2018 |

Andrade, J. B. D.; Pinheiro, H. L. C.; Andrade, M. V. A. S. d., The Formaldehyde and Acetaldehyde Content of Atmospheric Aerosol. Journal of the Brazilian Chemical Society 1995, 6, 287-290.

Odabasi, M.; Seyfioglu, R., Phase partitioning of atmospheric formaldehyde in a suburban atmosphere. Atmospheric Environment 2005, 39, (28), 5149-5156.

Klippel, W.; Warneck, P., The formaldehyde content of the atmospheric aerosol. Atmospheric Environment (1967) 1980, 14, (7), 809-818.

Shen, H. Q.; Chen, Z. M.; Li, H.; Qian, X.; Qin, X.; Shi, W. X., Gas-Particle Partitioning of Carbonyl Compounds in the Ambient Atmosphere. Environmental Science & Technology 2018, 52, (19), 10997-11006.

Figure 4: You need to indicate which of these panels are from observations and which are output from your model.

Response: Thank you for your kindly suggestions. We have revised the Figure 4 and rewritten the sentences.

[Figure]

Figure 4. Concentrations of $SO_2$, $SO_4^{2-}$, and HCHO observed at different pollution stages. The simulated HCHO (aq) and HMS (aq) were also presented. The particulate sulfur molar percentage was calculated as $[n(SO_4^{2-})+n(HMS)]/[n(SO_4^{2-})+n(HMS)+ n(SO_2)]$. In the box–whisker plots, the whiskers, boxes, and points indicate the 5th/95th, 25th/75th, 50th percentiles, and mean values.

Line 350: "through" is mis-spelled.

Response: Thanks. Corrected.

Line 354 and 359: HCHO (aq) and HMS were not directly observed, correct? You only calculate it with your model. Please don't use the word "observed" when you are talking about model output.

Response: Thank you for your good suggestions. We have revised the related sentences.

Section 3.5: My comment about model validation extends to this section as well, although it is less concerning given that your results are broadly consistent with other studies.

Response: Thank you for your kindly comments and suggestions. About the uncertainties of the model simulation results, the index of agreement (IOA) was used to judge the reliability of the model simulation results, as follows:

$$IOA = 1 - \frac{\sum_{i=1}^{n}(O_i - S_i)^2}{\sum_{i=1}^{n}(|O_i - \bar{O}| - |S_i - \bar{O}|)^2}$$

where $Si$ is simulated value, $Oi$ represents observed value, $\bar{O}$ is the average observed values, and n is the sample number. The IOA range is 0-1, and the higher the IOA value is, the better agreement between simulated and observed values is. In this study, the simulation results (the IOA is approximately 0.80) are reasonable, and the performance of the OBM-MCM model was acceptable.

Line 377: I'm unclear as to what you mean by input HCHO and non-input HCHO. As well as when, in the abstract for example, you say you "disabled the HCHO mechanism". Are you just not constraining your model to observed HCHO? Are you removing all reactions that produce or remove HCHO from the chemical mechanism? Please explain this more clearly.

Response: Thank you for your good comments.

To furtherly quantify the varieties in ROx chemistry and $O_3$ formation in response to HCHO chemistry, two parallel scenarios were conducted through the F0AM model. One scenario was run with all MCM mechanism defined as Input HCHO mechanism, and the other was run with the HCHO mechanism disabled in MCM mechanism defined as Non-input HCHO mechanism. Considering the numerous chemical formation reactions of HCHO, our study mainly disabled the HCHO loss pathways of HCHO photolysis, HCHO + OH, and HCHO + $NO_3$, which makes HCHO become a stable secondary pollutant that will not continue to react, thus the HCHO loss pathways were disabled in Non-input HCHO mechanism scenario. The differences in ROx levels and production pathways in the two model scenarios were analyzed to investigate the chain effect of HCHO on ROx cycling and $O_3$ formation. More details could be found in our previous studies (Liu et al., 2023).

These sentences have been added into the revised manuscript, as follows:

To furtherly quantify the effects of HCHO on ROx chemistry and $O_3$ formation, we disabled the loss pathways of HCHO photolysis, HCHO + OH, and HCHO + $NO_3$ in MCM mechanism, defined as Non-input HCHO mechanism (Liu et al., 2023). Meanwhile, the other scenario was run with all MCM mechanisms, defined as Input HCHO mechanism.

Liu, T., Lin, Y., Chen, J.*, Chen, G., Yang, C., Xu, L., Li, M., Fan, X., Zhang, F., and Hong, Y.* Pollution mechanisms and photochemical effects of atmospheric HCHO in a coastal city of southeast China. *Science of the Total Environment*. 2023, 859:160210